# Glial Gap Junction Pathology in the Spinal Cord of the 5xFAD Mouse Model of Early-Onset Alzheimer’s Disease

**DOI:** 10.3390/ijms232415597

**Published:** 2022-12-09

**Authors:** Maria Pechlivanidou, Ioanna Kousiappa, Stella Angeli, Irene Sargiannidou, Andreas M. Koupparis, Savvas S. Papacostas, Kleopas A. Kleopa

**Affiliations:** 1Neurobiology Department, The Cyprus Institute of Neurology and Genetics, Nicosia 2371, Cyprus; 2Medical School, University of Nicosia, Nicosia 2414, Cyprus; 3Neuroscience Department, The Cyprus Institute of Neurology and Genetics, Nicosia 2371, Cyprus; 4Epilepsy Centre, The Cyprus Institute of Neurology and Genetics, Nicosia 2371, Cyprus; 5Dementia and Cognitive Disorders Centre, The Cyprus Institute of Neurology and Genetics, Nicosia 2371, Cyprus; 6Center for Neuromuscular Disorders, The Cyprus Institute of Neurology and Genetics, Nicosia 2371, Cyprus; 7Center for Multiple Sclerosis and Related Disorders, The Cyprus Institute of Neurology and Genetics, Nicosia 2371, Cyprus

**Keywords:** Alzheimer’s disease, gap junctions, neurodegeneration, 5xFAD, spinal cord, motor deficits

## Abstract

Gap junctions (GJs) are specialized transmembrane channels assembled by two hemi-channels of six connexin (Cx) proteins that facilitate neuroglial crosstalk in the central nervous system (CNS). Previous studies confirmed the crucial role of glial GJs in neurodegenerative disorders with dementia or motor dysfunction including Alzheimer’s disease (AD). The aim of this study was to examine the alterations in astrocyte and related oligodendrocyte GJs in association with Aβ plaques in the spinal cord of the 5xFAD mouse model of AD. Our analysis revealed abundant Aβ plaque deposition, activated microglia, and astrogliosis in 12-month-old (12M) 5xFAD mice, with significant impairment of motor performance starting from 3-months (3M) of age. Additionally, 12M 5xFAD mice displayed increased immunoreactivity of astroglial Cx43 and Cx30 surrounding Aβ plaques and higher protein levels, indicating upregulated astrocyte-to-astrocyte GJ connectivity. In addition, they demonstrated increased numbers of mature CC1-positive and precursor oligodendrocytes (OPCs) with higher immunoreactivity of Cx47-positive GJs in individual cells. Moreover, total Cx47 protein levels were significantly elevated in 12M 5xFAD, reflecting increased oligodendrocyte-to-oligodendrocyte Cx47–Cx47 GJ connectivity. In contrast, we observed a marked reduction in Cx32 protein levels in 12M 5xFAD spinal cords compared with controls, while qRT-PCR analysis revealed a significant upregulation in *Cx32* mRNA levels. Finally, myelin deficits were found focally in the areas occupied by Aβ plaques, whereas axons themselves remained preserved. Overall, our data provide novel insights into the altered glial GJ expression in the spinal cord of the 5xFAD model of AD and the implicated role of GJ pathology in neurodegeneration. Further investigation to understand the functional consequences of these extensive alterations in oligodendrocyte–astrocyte (O/A) GJ connectivity is warranted.

## 1. Introduction

In the central nervous system (CNS), neurons and glia are organized into complex networks that interact to form pan glial syncytia [1,2,3]. CNS glia, referring mainly to microglia, astrocytes, and oligodendrocytes, are responsible for maintaining overall homeostasis and play a crucial role in mediating cell-to-cell communication between neurons and glia cells [1,2,4]. Inter- and intracellular communication is supported by gap junctions (GJs), which are specialized transmembrane channels composed of two opposing hemichannels (HCs) known as connexons, between the plasma membrane of adjacent cells [5,6,7]. Each connexon is formed by six connexin proteins that can be organized either into the homomeric or heteromeric type of HCs [3,5,8]. GJ channels enable the flow of molecules smaller than 1000 Da, including ions, ATP, glutamate, Ca^2+^, and cAMP, which mediate electrical coupling, various signaling events, growth control, and cellular differentiation [1,2,4]. In particular, GJs facilitate the functional crosstalk between astrocytes and oligodendrocytes through the intercellular coupling of astrocytes to astrocytes (A:A), oligodendrocytes to oligodendrocytes (O:O), and astrocytes to oligodendrocytes (A:O) [5,7,9]. Inter-astrocytic A/A GJ coupling is mediated via homotypic Cx43/Cx43 and Cx30/Cx30 channels, with the latter found mostly in grey matter (GM) [2,3,9,10,11]. Oligodendrocytic Cx47 facilitates the majority of intercellular O/O coupling via homotypic Cx47/Cx47 channels, and it also mediates O/A coupling via heterotypic Cx47/Cx43 channels [2,9,12]. In humans and mice, Cx32 is also expressed at oligodendrocyte cell bodies and processes in GM and white matter (WM), and it participates in O/O coupling via Cx32/Cx32 and in O/A GJs coupled with its astrocytic partner Cx30 [2,5,6,13,14,15,16]. In addition, Cx32 forms reflexive GJs within the myelin sheath especially along large myelinated fibers found in long tracks of the spinal cord and other CNS WM areas [5,9,10,16].

Alzheimer’s disease (AD) is the most prominent and irreversible cause of dementia in individuals aged 65 and above, and it is characterized not only by cognitive decline but also by progressive loss of motor function [17]. Remarkably, motor impairment appears at early stages of the disease and precedes the expression of dementia symptoms [18,19,20]. Microscopically, the disease is characterized by the extracellular deposition of beta-amyloid (Aβ) in senile plaques and intracellular hyperphosphorylated tau proteins that aggregate into neurofibrillary tangles [21,22,23].

Glial GJs have been implicated in AD neuroinflammation and neurodegeneration. In post-mortem AD brain samples and in murine AD models, astroglial Cx30 and particularly Cx43 in reactive astrocytes accumulate significantly around β-amyloid plaques [24,25,26], and overactivation of Cx43 hemichannels has been linked to excessive ATP and glutamate release with toxic consequences for neurons [26,27,28,29,30]. However, the pathological role of astroglial and oligodendroglial GJs in the spinal cord of AD models has not yet been investigated.

In the present study, we investigated for the first time the expression profiles of astroglial GJ proteins Cx43/Cx30 and oligodendroglial Cx47/Cx32 in relation to AD pathology in the spinal cord of the 5xFAD AD mouse model. We detected Aβ-induced severe astrogliosis and increased immunoreactivity of Cx43 and Cx30 GJs around Aβ plaques. Immunoblot analysis confirmed increased Cx43 protein levels and strongly implicated their involvement in AD progression. Furthermore, we showed for the first time increased immunoreactivity of the oligodendroglial Cx47-formed GJs plaques on the cell bodies and processes of mature oligodendrocytes associated with elevated mRNA and Cx47 protein levels, suggesting an increased Cx47–Cx47 O/O GJs connectivity. In contrast, the levels of the second oligodendroglial GJ protein Cx32 were reduced along with marked loss of intra-myelin reflexive Cx32-formed GJs in large myelinated fibers of the white matter, which possibly contributed to myelin disruption.

## 2. Results

### 2.1. Beta-Amyloid Pathology within the 5xFAD Spinal Cord

The distribution and concentration of Aβ plaques in the 2nd–3rd cervical, 2nd–3rd thoracic and 2nd–3rd lumbar spinal sections of 3M and 12M 5xFAD (Figure 1A,B) and wild-type (WT) mice were assessed by measuring and quantifying the percentage of the total grey matter (GM) area that was covered by 6E10 immunoreactivity using Image-J. In the GM, most of the plaques were localized around the intermediate GM zone, and fewer were found at dorsal and ventral horn areas. Additionally, 12M 5xFAD mice showed abundant extracellular Aβ plaque deposition in the GM of the aforementioned spinal cord segments, which appeared at three months and exhibited a time-dependent increase. The cervical sections had a higher number of Aβ plaques compared to the thoracic and lumbar sections in 12M transgenic mice (Figure 1C). In addition, 6E10-positive plaques were localized at the corticospinal, rubrospinal, reticulospinal, and lateral vestibulospinal descending tracts of the white matter (WM) [31]. The total numbers of Aβ plaques measured were: 160 plaques in 3M 5xFAD cervical, 110 plaques in 3M 5xFAD lumbar, 444 plaques in 12M 5xFAD cervical, and 270 plaques in 12M 5xFAD lumbar. In contrast, WT mice exhibited no Aβ deposits at any age.

### 2.2. Neuronal Loss in the Spinal Cord of 12M 5xFAD Mice

The degree of amyloid pathology in particular brain regions of the 5xFAD model has been previously correlated to the degree of neuronal degeneration [32]. Thus, we examined whether AD pathology could affect the lower motor neurons of the spinal cord. NeuN^+^ neurons were counted manually with ImageJ in the GM of cervical (C2–C3) and lumbar (L2–L3) hemi-sections of 3M 5xFAD, 12M 5xFAD, and WT mice. The number of mature NeuN^+^ neurons in cervical and lumbar GM in 3M 5xFAD (cervical: 543 ± 33.35, lumbar: 475 ± 14.83 NeuN^+^ cells) was similar to those in WT mice of the same age (cervical: 546 ± 32.70, lumbar: 474 ± 6.603 NeuN^+^ cells) (Figure 2A,B). However, the number of mature NeuN^+^ neurons in the cervical segment was significantly lower in 12M 5xFAD mice (cervical: 439 ± 18.66, lumbar: 420 ± 7.360 NeuN^+^ cells) compared with age-matched WT littermates (cervical: 494 ± 30.30, lumbar: 450 ± 10.59 NeuN^+^ cells) (Figure 2A,B). Interestingly, 12M WT mice exhibited a small but significant neuronal loss in comparison to the 3M WT animals at both spinal cord levels. Taken together, although a small age-related neuronal loss occurred also in WT mice, a marked neuronal loss occurred in the context of AD pathology, which was associated with progressive Aβ deposition in the GM of 5xFAD mice.

### 2.3. Disease-Related Decline in Motor Function in 5xFAD Mice

The rotarod test was performed to assess motor learning and coordination in 5xFAD mice. At the age of three months, transgenic and WT mice exhibited a similar motor behavior, as they fell from the rotarod at similar time points. However, 12M 5xFAD mice showed a deterioration of motor performance, and they fell from the rotarod significantly faster than the rest of the mouse groups across all trials (Figure 3A). Furthermore, while 3M 5xFAD and WT mice did not miss steps during the performance of the foot-slip test, 12M 5xFAD mice exhibited significantly more missed steps in comparison with their age-matched WT littermates and with 3M 5xFAD mice (Figure 3B).

The sensory-motor functionality of mice was further examined by performing the grip strength test, which measures the ability of a mouse to hold a grip with its paws and allows the resulting pulling force to be calculated. 12M 5xFAD mice displayed significantly lower grip strength compared with their WT littermates and with 3M 5xFAD mice (Figure 3C). Moreover, a few 12M transgenic mice failed to hold on to the wire of the grip strength meter or they released it significantly faster than WT mice of the same age.

The footprint test was performed to evaluate gait and coordination. 3M 5xFAD mice did not differ in stride length and stride width from age-matched WT mice (Figure 3D,E), while 12M 5xFAD mice displayed a significantly shorter mean stride length and mean stride width (Figure 3F,G). These data suggest that the older 5xFAD mice showed evidence of impaired gait with dysfunctional locomotion in comparison with the younger transgenic mice, which could move with higher speed during the behavioral test.

Taken together, the results of behavioral analysis revealed marked deficits in motor performance and co-ordination in 5xFAD mice, which were progressive and more severe in older animals, correlating with the accumulating AD-related pathology in the brain and spinal cord.

### 2.4. Severe Astrogliosis in the Spinal Cord of 12M 5xFAD Mice

Neuroinflammation and glia activation are one of the major pathological hallmarks in neurodegenerative diseases. In the 5xFAD brain, particularly in the cortex and thalamus, reactive astrocytes and microglia that have been shown to surround Aβ plaques release pro-inflammatory mediators contributing further to pathology progression [27,33,34,35]. However, the degree of gliosis and neuroinflammation in the 5xFAD spinal cord has not been investigated so far. Therefore, we stained for GFAP and Iba1 markers to assess the levels of GFAP-positive reactive and proliferative astrocytes and Iba1-positive activated microglia, respectively (Figure 4A,D). The percentage of total GM area occupied by the immunofluorescence of GFAP^+^ reactive astrocytes in the GM of both cervical and lumbar sections was higher in 3M 5xFAD (cervical: 6.47 ± 2.808%, lumbar: 7.51 ± 2.792%) compared with WT mice of the same age (cervical: 1.59 ± 1.063%, lumbar: 1.64 ± 1.103%). In addition, a significantly higher degree of astrogliosis was demonstrated by a higher percentage of GFAP^+^ reactive astrocytes, a higher number with hypertrophic processes, and enlarged cell bodies in 12M 5xFAD (cervical: 13.26 ± 3.311%, lumbar: 11.35 ± 4.613%) compared with 3M 5xFAD mice (Figure 4B,E). In contrast, 12M WT mice (cervical: 1.61 ± 1.124%, lumbar: 1.53 ±1.328%), showed a significantly lower percentage of GFAP^+^ astrocytes compared with 5xFAD mice. Likewise, quantification of Iba1 immunoreactivity confirmed that microglia were significantly activated in 12M 5xFAD mice, as they appeared enlarged, and the percentage of the total GM area covered by Iba1+ immunofluorescence was higher (cervical: 24.79 ± 8.153%, lumbar: 20.84 ± 5.688%) compared with their age-matched WT littermates (cervical: 1.63 ± 1.570%, lumbar: 2.04 ± 1.440%) and with 3M 5xFAD mice (cervical: 12.14 ± 3.528%, lumbar: 12.13 ± 3.528%) (Figure 4C,F).

These data suggest that astrocyte and microglia activation followed a time-dependent and Aβ-dependent progressive increase in the cervical and lumbar spinal cord GM of 12M 5xFAD mice.

### 2.5. Elevation of Astroglia Cx43 and Cx30 around B-Amyloid Plaques within the 5xFAD Spinal Cord

Given the severe degree of astrogliosis in the spinal cord GM of 5xFAD mice surrounding Aβ plaques, we next examined the expression of the major astrocytic GJ proteins Cx43 and Cx30 in cervical and lumbar samples from 3M and 12M 5xFAD and WT mice. Double immunostaining for Aβ/Cx43 and Aβ/Cx30 was performed, and the fluorescence intensity at the perimeter of each individual Aβ plaque was assessed. In addition, the fluorescence intensity in areas away from plaques was calculated for both connexins and was compared with the corresponding areas in age-matched WT mice.

In the GM of both cervical and lumbar segments, the immunoreactivity of Cx43 in the microenvironment of the plaques was significantly higher in 12M 5xFAD mice (cervical: 1,622,517 ± 271,363, lumbar: 1,491,106 ± 302,830 a.u.) compared both with the areas away from plaques measured in the same field (cervical: 816,683 ± 264,405, lumbar: 783,377 ± 263,587 a.u.) and with the corresponding areas in 12M WT controls (cervical: 618,487 ± 159,181, lumbar: 783,999 ± 62,638 a.u.) (Figure 5A,B). These data confirmed that Cx43 expression was specifically increased only in the environment of the plaques and not far away from them. Furthermore, the mean fluorescent intensity of Cx43 around the Aβ plaques was significantly higher and more intense in comparison with the corresponding mean value obtained from the 3M 5xFAD mice (Figure 5E,F). The immunoreactivity levels of Cx43 in 3M 5xFAD mice around Aβ plaques (cervical: 821,546 ± 512,311, lumbar: 661,524 ± 290,867 a.u.) was similar to the expression of Cx43 in areas away from plaques (cervical: 553,285 ± 360,706, lumbar: 529,400 ± 342,729 a.u.) and in corresponding areas of age-matched WT mice (cervical: 321,292 ± 114,483, lumbar: 420,003 ± 231,333 a.u.).

The immunoreactivity of Cx30, which is mostly expressed in the GM [10,11,36,37], was highly increased around Aβ plaques in both spinal cord segments, similar to Cx43, in 12M 5xFAD (cervical: 1,988,372 ± 438,678, lumbar: 1,529,400 ± 209,281 a.u.) compared with the 3M 5xFAD mice (cervical: 438,729 ± 298,140, lumbar: 774,450 ± 121,995 a.u.) (Figure 5C,D). Similarly, in both spinal cord samples, the expression of Cx30 in 12M transgenic mice was significantly higher in comparison with the areas away from plaques (cervical: 799,265 ± 150,400, lumbar: 952,814 ± 196,787 a.u.) and with the corresponding areas in age-matched WT mice (cervical: 801,947 ± 67,732, lumbar: 583,255 ± 74,894 a.u.) (Figure 5G,H). In 3M transgenic mice, Cx30 immunoreactivity levels (cervical: 738,729 ± 298,140, lumbar: 474,450 ± 121,995 a.u.) were not significantly different from those obtained in areas away from plaques (cervical: 532,782 ± 210,104, lumbar: 659,640 ± 211,139 a.u.) and in corresponding areas of WT littermates (cervical: 647,120 ± 121,735, lumbar: 523,422 ± 272,101 a.u.) (Figure 5G,H). Thus, both Cx43 and Cx30 are highly increased in the microenvironment of the Aβ plaques in older 5xFAD mice as a result of the progression of AD-related astrogliosis, indicating the establishment of increased A/A GJ connectivity.

### 2.6. Increased Cx43 Protein Levels in the Spinal Cord of 12M 5xFAD Mice

Following the investigation of the regional expression of Cx43 and Cx30 in the microenvironment of the Aβ plaques located in the GM of the spinal cord, we wanted to examine the overall expression levels in samples from cervical and lumbar tissues.

In the cervical spinal cord, mRNA levels of *Cx43* were significantly higher in 12M 5xFAD mice compared with the 3M transgenic group (Figure 6A). Interestingly, there was no significant difference in the mRNA levels between the 5xFAD and WT mice of 12 months of age, despite a significant difference between the 3M 5xFAD mice and the WT mice of the same age group. Furthermore, in lumbar spinal cord samples, *Cx43* mRNA levels did not differ in 5xFAD and WT mice at any age (Figure 6C). *Cx43* mRNA levels were lower in the 12M compared to the 3M group, but it was not statistically significant. Immunoblot analysis of Cx43 revealed that the 12M 5xFAD group displayed a significant increase at the protein level in both spinal cord samples compared with their age-matched WT littermates, while at 3 months, Cx43 levels appeared elevated only in cervical samples without statistical significance (Figure 6B,D). A significant upregulation of Cx43 was found from 3 to 12 months of age in 5xFAD mice but only at the cervical spinal cord level (Figure 6B). These findings could result from the higher Aβ plaque load in the cervical compared with the lumbar section. The results from the Cx43 immunoblot analysis line up with the immunohistochemistry data, indicating increased formation of a diffuse A/A connectivity specifically around the microenvironment of plaques.

On the other hand, *Cx30* mRNA levels displayed only a very small (but significant) upregulation in the cervical tissue of the 12M mice, while the 3M group showed a non-significant reduction (Figure 6E). The mRNA levels of *Cx30* were the same between transgenic and WT mice of all age groups. Remarkably, lumbar spinal cord *Cx30* mRNA levels revealed a significant age-related decrease in 12M 5xFAD mice compared with their WT littermates and with 3M 5xFAD mice (Figure 6G). Furthermore, immunoblot analysis showed that Cx30 protein levels were slightly higher in the 12M 5xFAD mice compared with the control group and with 3M 5xFAD mice (Figure 6F,H), although this was not statistically significant. The Cx30 immunoblot results could also be explained by the fact that Cx30 is mainly restricted to GM; therefore, immunohistochemistry analysis revealed a more specific pattern of Cx30 expression by the higher immunoreactivity only around the core of beta-amyloid plaques.

Taken together, the results of imaging, molecular, and biochemical analysis of astrocyte connexins confirmed an age-dependent increase in A/A GJ formation in 5xFAD spinal cords mostly by Cx43 but also by Cx30 focally, with the latter being restricted to GM.

### 2.7. Increased Immunoreactivity of Oligodendroglia Cx47 within the GM of the 5xFAD Spinal Cord

Following the altered expression of astrocytic connexins, we examined their oligodendroglial partners to investigate any possible disruption of O/A GJs in the GM of the 5xFAD spinal cord. Double immunostaining was performed using the CC1 marker to stain mature oligodendrocytes and the anti-Cx47 antibody (Figure 7A,B). The fluorescence intensity of Cx47-positive GJ plaques was estimated in mature oligodendrocytes located in the GM including the deposits of the Aβ senile plaques. Cx47-immunoreactive GJ plaques were localized mainly in the cell bodies of CC1^+^ mature oligodendrocytes and in their proximal processes (Figure 7A,B). Quantification of the fluorescent intensity of Cx47-positive GJ plaques revealed a significantly higher degree of Cx47 immunoreactivity in cervical and lumbar GM of 12M 5xFAD mice (cervical: 475,226 ± 103,733, lumbar: 401,346 ± 87,544 a.u.) compared with their WT littermates (cervical: 276,742 ± 51,791, lumbar: 253,906 ± 71,086 a.u.) or with the 3M 5xFAD group (cervical: 244,428 ± 31,677, lumbar: 244,428 ± 31,677 a.u.) (Figure 7C,D). The 3M 5xFAD group showed similar levels of Cx47 immunoreactivity compared to their WT littermates (cervical: 212,658 ± 34,573, lumbar: 264,332 ± 20,012 a.u.) (Figure 7C,D). Additionally, we manually counted the number of CC1^+^ mature oligodendrocytes expressing Cx47-positive GJs and found them to be higher in the cervical and lumbar GM of the 12M but not of the 3M 5xFAD spinal cord compared with WT littermates (Figure 7E,F).

### 2.8. Increased Numbers of Mature Oligodendrocytes and Oligodendrocyte Precursors (OPCs) Expressing Cx47 within the GM of the 5xFAD Spinal Cord

Furthermore, the effect of Aβ plaques and astrogliosis on the survival and function of cell populations that originate from the oligodendrocyte lineage was investigated within the GM of the 5xFAD spinal cord. Double immunostaining was performed using the Olig2 marker for oligodendrocyte precursor cells (OPCs) and anti-Cx47 antibody (Figure 8A,B). We followed the same methodology to analyze the population of OPCs by manually counting the Olig2^+^, Cx47-positive GJs in GM. Interestingly, in both cervical and lumbar GM the mean number of Olig2^+^ OPCs that were Cx47-positive did not differ between 3M 5xFAD and WT mice, but they were significantly higher in 12M 5xFAD mice compared with their WT littermates (Figure 8C,D). These data indicate that the presence of Aβ plaques in combination with astrogliosis lead to age-dependent induction of mature oligodendrocytes and OPCs with Cx47-positive puncta located within the GM of 5xFAD mice.

### 2.9. Increased Cx47 Protein Levels in the Spinal Cord of 12M 5xFAD Mice

Quantitative qPCR analysis demonstrated that the mRNA levels of *Cx47* were significantly lower in cervical samples of 12M compared with the 3M 5xFAD mice (Figure 9A), while no significant differences were found between 5xFAD mice and their WT littermates at both age groups in lumbar spinal cord samples (Figure 9C). However, at the protein level, Cx47 was increased in 12M 5xFAD compared with their WT littermates in both cervical and lumbar spinal cord tissue samples (Figure 9B,D). This Cx47 increase was not present in 3M mice groups. Thus, even though the mRNA levels of *Cx47* remained similar between the different age groups, Cx47 protein levels were increased in older 5xFAD mice in comparison with 3M 5xFAD and WT groups. In addition, these results confirmed the immunostaining findings that demonstrated increased Cx47 GJ plaque formation and O/O GJ connectivity in the GM of 12M 5xFAD mice with accumulated Aβ plaque burden.

### 2.10. Partial Disruption of O/A GJ Connectivity in the GM of the 5xFAD Spinal Cord

Given these alterations in the expression of both Cx43 and Cx47 that are responsible for forming the majority of A/O GJs, we performed double immunostaining for Cx47 and Cx43 to investigate the co-localization status of O/A GJ connectivity within the GM of the 5xFAD spinal cord. In cervical and lumbar GM of 3M 5xFAD and 12M WT mice, Cx43 immunoreactivity mostly overlapped with that of Cx47, surrounding oligodendrocytes, indicating a normal degree of heterotypic O/A GJ channel formation (Figure 10B,C,E,F). However, 12M 5xFAD mice displayed a partial disorganization of O/A GJ expression in both spinal cord sections, showing more Cx47-positive GJ plaques that do not overlap with Cx43 (Figure 10A–D). These findings suggest a widespread disturbance of O/A connectivity, as Cx47-positive GJs seem to be more numerous than Cx43 GJs at the sites of their connection, while non-overlapping Cx47 and Cx43 GJ plaques are increased.

### 2.11. Decreased Cx32 Protein Levels in the Spinal Cord of 12M 5xFAD Mice

In addition to Cx47, we also studied possible alterations of the second GJ protein in oligodendrocytes: Cx32. Real-time PCR analysis showed significantly increased *Cx32* mRNA levels in the cervical and lumbar samples of 12M 5xFAD compared with the 3M 5xFAD mice as well as with their WT littermates (Figure 11A,C). In contrast, Cx32 protein expression levels assessed by immunoblot were reduced in the cervical and lumbar spinal cord tissue samples of 12M 5xFAD mice compared with their WT littermates (Figure 11B,D). Quantification of normalized band optic density (OD) was not performed because the low volume of lysates did not allow repeated Cx32 immunoblots for the statistical analysis. In keeping with these results, Immunofluorescence staining showed reduced expression of Cx32 in the cervical and lumbar white matter ventral funiculus (Appendix A). Overall, these data suggest an age-dependent reduction in Cx32 protein levels in the spinal cord of 12M 5xFAD mice and a likely reactive elevation of *Cx32* mRNA levels that may reflect a compensatory molecular mechanism.

### 2.12. Disruption of Myelin Microstructure and Axonal Preservation at the Environment of the Aβ Senile Plaques

As Cx32 is mainly expressed in myelinated fibers, we further performed double Aβ/PLP and Aβ/NF-H immunostaining to examine the integrity of myelin and axons in the local environment surrounding amyloid plaques (Appendix A). The PLP marker was used to stain myelin proteolipid protein, a component of CNS compact myelin, and NF-H serves as a marker of axonal heavy neurofilaments. In contrast to the increased number of GM oligodendrocyte populations in 12M 5xFAD mice, we observed local myelin deficits around amyloid plaques, as the PLP immunoreactivity was very weak in those areas (Appendix A. However, immunoblot analysis led to non-significant results when comparing 5xFAD and WT mice of both age groups. Nevertheless, myelin basic protein (MBP) levels appeared to be reduced in the cervical and lumbar samples of 12M 5xFAD mice as opposed to controls (Appendix A.

Additionally, the immunoreactivity of the NF-H marker at the center of the senile plaques in cervical and lumbar GM remained similar between transgenic and WT mice of both age groups (Appendix A). Moreover, NF-H protein levels were similar across all transgenic and WT groups. Despite these non-significant findings, immunoblot analysis indicated reduced neurofilament levels in 12M 5xFAD mice (Appendix A). These results suggest that axons were generally preserved around the microenvironment of the plaques even though the myelin was partially deficient.

## 3. Discussion

The main goal of this study was to evaluate possible alternations in the expression of glial GJ proteins and their association with the progressive accumulation of Aβ-plaque pathology within the spinal cord of the 5xFAD model of early-onset AD. We present novel findings of Aβ-induced increased expression of astrocytic Cx43 and its oligodendrocytic partner Cx47 accompanied by proliferation of oligodendrocyte lineage cells. We also observed reduced expression of the second oligodendrocyte GJ protein Cx32 in the cervical and lumbar GM of the 12M 5xFAD model. We propose an astrogliosis-driven shift in A/A GJ connectivity, implying an overactivation of Cx43 hemichannels and upregulated O/O GJ coupling supported by the augmentation of mature and precursor oligodendrocytes. Widespread disturbance of O/A GJ communication, including the partial loss of Cx43–Cx47 GJ pairing and a possible selective loss of intra-myelin GJ channels, could favor a continuous generation of an excitotoxic environment leading to neuronal stress, disrupted myelin/axonal homeostasis, and neurodegeneration.

We chose to focus our investigation on the 5xFAD model, an excellent and well-studied murine model of familial AD that reproduces several aspects of human disease [38]. From our evaluation in the spinal cord, it was evident that the extracellular amyloid deposits appeared at three months of age, and the cervical GM segment of 12M 5xFAD mice displayed the heaviest Aβ plaque load compared with the rest of the spinal cord sections, which is in keeping with previous reports [31]. Under physiological conditions, the cervical section of the spinal cord has a higher number of lower motor neurons than thoracic and lumbar sections; therefore, the synaptic activity is expected to be more intense and could explain the higher rate of released Aβ plaque deposits [31]. Furthermore, extracellular Aβ plaques were also located in columns of the white matter (WM), which in rodents’ spinal cord correspond to the lateral corticospinal tract (CST), the rubrospinal, reticulospinal, and lateral vestibulospinal tracts [31]. From the combination of our findings and results from a recent study [39], it could be accepted that despite the differences in anatomy, the rate of the progressive accumulation of amyloid senile plaques in the brain and spinal cord follows a similar chronological and neuropathological profile [38]. Interestingly, Yuan et al. proposed that Aβ peptides leak out from the CST and become localized within the spinal cord, which could serve as a molecular mechanism for the simultaneous development and presence of plaque-related AD pathology in the brain and spinal cord [40,41]. This proposal was experimentally confirmed upon the depletion of the sensorimotor cortex and the immediate release of intraneuronal Aβ peptides in the dorsal GM and WM columns in the TgCRND8 mouse spinal cord [40,41]. Probably a similar pathway could serve as an origin for the release and localization of Aβ peptides in the 5xFAD spinal cord, even though the mechanism of the plaque-related transportation and leakage remains unclear.

Additionally, neuronal loss in the 5xFAD model begins around six months of age at various cortical regions but becomes more intense at layer V of the cortex and in the basal forebrain, where the cholinergic neurons are found to be statistically reduced at nine months of age compared to controls [32,42,43,44]. Here, we also detected significant neuronal loss in 12M 5xFAD mice in contrast to previous reports indicating that the amyloid plaques were not toxic to the lower motor neurons of the ventral horn and failed to find neuronal loss in six-month-old 5xFAD mice [31]. It remains controversial whether the 5xFAD mouse model suffers neuronal death in the spinal cord too; therefore, further experiments should be performed investigating the association of beta-amyloid toxicity with neuronal and synaptic loss.

Many of the major neuropathological hallmarks of AD, including cognitive dysfunction and memory deficits, have been successfully represented by various AD transgenic mouse models. We examined the motor learning behavior and function in the 5xFAD model by assessing their coordination and muscle strength with the rotarod test followed by the foot-slip and hind limb grip strength analysis, and we assessed their posture by performing the footprint analysis. From our behavioral analysis, we demonstrated that 12M 5xFAD mice had the lowest motor performance score, reduced motor speed, balance, and impaired gait, which is in line with findings from similar studies conducted in the 5xFAD model, APP/PS1K1 model, and other AD models [44,45,46,47]. Motor disabilities are not clinical features only in APP-based AD transgenic models but also in PS19 mice, a tauopathy AD model with observed motor decline and walking impairment at nine months of age [48]. The motor impairments that we detected in the 5xFAD model reflect the analogous clinical deficits in patients with AD, coinciding with the degree of amyloid-plaque load and AD-induced inflammation in the spinal cord [46,49]. These data verify the benefits of 5xFAD model utilization in AD-linked research investigating sensorimotor dysfunctional mechanisms and testing candidate therapeutics, as it is characterized by both memory and motor decline, recapitulating the major clinical and pathological symptoms of individuals with AD and/or mild cognitive impairments (MCI) [19,20,44].

In AD, neurons could release soluble neurotoxic Aβ peptides to the extracellular space, which in turn stimulate microglial cells to produce proinflammatory mediators, such as tumor necrosis factor-alpha (TNF-a) and reactive oxygen species (ROS), leading to the activation of astrocytes around the area of the extracellular beta-amyloid plaques [50,51,52]. Reactive astrocytes respond by releasing chemokines and cytokines such as interleukin-1-beta (IL-1β), which maintain the vicious Aβ-generation cycle with neuroinflammatory and neurodegenerative consequences [3,51,53]. In our study, we detected severe astrogliosis reflected by the higher number of immunoreactive GFAP-positive activated astrocytes with thick processes and increased density as well as Iba1-positive microglia around Aβ plaques in the GM of the 12M 5xFAD mice compared with the WT controls. Our results from the spinal cord analysis confirmed an association between progressive accumulation of senile amyloid plaques and the severity of reactive astrogliosis. These findings are in accordance with other publications that demonstrate the proliferation of GFAP-activated astrocytes starting early in the brain of the 5xFAD model in parallel with the aggressive accumulation of Aβ deposits [27,35,39]. In addition, glial activation is a pathological hallmark in tauopathy-associated AD models, such as in PS19, in which reactive astrogliosis and microgliosis were prominent in brain regions and occurred in parallel to tau aggregation [48]. Nonetheless, neurotoxic astrogliosis has also been detected in other CNS disorders without the presence of AD-related Aβ pathology, including Multiple Sclerosis (MS) with GM and WM inflammatory demyelination [1,54]. Overall, inflammatory gliosis is prominent in AD pathology and may lead to the dysregulation of glutamate transporters, which may promote enhanced excitotoxicity that causes focal neuronal damage and death, resulting in cognitive and motor decline [51,55].

Cx43 and Cx30 are the dominant connexins expressed by astrocytes, and their expression patterns are region-specific [3,25,56]. We provide evidence that both astroglial connexins were affected by Aβ-induced inflammation and gliosis in the cervical and lumbar spinal cord segments of 5xFAD mice. Both connexins exhibited higher immunoreactivity at the Aβ peri-plaque area in the 12M 5xFAD mice, which displayed the heaviest plaque accumulation and the most severe degree of astrogliosis and microgliosis. Immunoblot analysis confirmed increased protein levels of Cx43 in 12M 5xFAD. However, Cx30 showed no significant changes, likely because it is restricted to the GM [10] and subtle alterations in the vicinity of the senile Aβ plaques could not be detected by immunoblot of the whole cord tissue. Nevertheless, the involvement of Cx43 in AD pathology is more established than Cx30, as has already demonstrated by similar findings obtained in the AD murine brain [25,26,39].

Upregulated A/A GJ connectivity mediated by the overexpression of Cx43 GJs upon astrogliosis in neurodegenerative diseases has been linked to neuronal excitotoxicity via excessive calcium signaling and diminished homotypic Cx43–Cx43 GJ coupling, which leads to the overactivation of Cx43 hemichannels [57]. Almad et al. provided evidence of upregulated Cx43 GJs and Cx43 hemichannels in cellular membranes of iPSC-derived astrocytes (hiPSC-A) from familial and sporadic case of ALS, which stresses the importance of novel GJ-modifying therapies for AD, ALS, and inflammatory neurodegenerative disorders [58]. In AD specifically, Cx43 hemichannels of GFAP-positive activated astrocytes in close proximity to the Aβ plaques are triggered by microglia-produced TNF-a, IFN-γ, IL-1B, and they release excessive ATP and gliotransmitters including glutamate [26,30,59,60,61].

Along with these astrocyte connexin changes, we demonstrated for the first time increased immunoreactivity of the Cx47-positive GJ plaques expressed on cell bodies and proximal processes of CC1-positive mature oligodendrocytes adjacent to Aβ plaques in the cervical and lumbar GM of 12M 5xFAD mice. The increase in Cx47 immunoreactivity was higher in 12M than 3M 5xFAD mice and was more significant in the cervical than the lumbar GM. Contrary to our study, in the ventral posteromedial (VPM) nucleus of the thalamus and the primary/secondary motor (MOp/Mos) areas of the cortical layer V of the brain in nine-month-old 5xFAD mice, a reduction in Cx47 immunoreactivity in mature oligodendrocytes near Aβ plaques was noted, which was attributed to severe astrogliosis and AD-induced pathology [1,39,62]. Moreover, the number of mature oligodendrocytes and their precursors was also increased in 12M 5xFAD mice, which is consistent with Cx47 upregulation and increased Cx47–Cx47 GJ coupling. We believe that Aβ-induced reactive astrocytes may trigger the proliferation of OPCs to restore myelin integrity, as also observed in the normal appearing gray matter in MS patients [54]. However, in the cortex and thalamus of nine-month-old 5xFAD mice, mature oligodendrocytes and OPCs were depleted, which was linked to O/O GJ loss with neurodegenerative effects [1,39]. These divergent results may be explained by the fact that the CC1^+^ oligodendrocytes that were upregulated in the 5xFAD spinal cord likely do not share the same cell lineage identity as the CC1^+^ population that was found depleted in the brain of the 5xFAD model and could originate from heterogeneous polydendrocytes with differentiation capacity [63,64]. Interestingly, another study reported rapid proliferation of oligodendrocytes from a distinct type of TrkB oligodendrocyte lineage (OL), suggesting that glial cells in the spinal cord encounter different communication dynamics than those in the brain, and in response to peripheral traumatic injuries, they could be triggered to multiply [65]. These compelling findings indicate that in the same AD model, the oligodendroglial Gx47 GJs reflect a different and complex region-specific behavior that might imply the involvement of different regulatory pathways and signaling molecules driven by AD pathology.

The significance of altered Cx47 expression is highlighted by studies in MS models that showed loss of Cx47–Cx43 GJs and increased A/A GJ connectivity in the setting of astrogliosis, could represent a mechanism of the widespread pathology and demyelination [4,66,67,68]. Further investigation is warranted to understand the implication of altered Cx47 GJ expression in AD and in other acquired CNS disorders. Thus, the existence of Cx47–Cx43 channels with disproportional GJ coupling may drive AD progression through loss of restrictive ionic permeability and directional coupling as well as disordered spatial buffering of K^+^ ions, a mechanism demonstrated through the expression of the Cx47 P90S mutation [67,69].

In contrast to increased astroglial connexins and Cx47, we found a marked loss of oligodendroglial Cx32 within the spinal cord of 12M 5xFAD mice, although we did not perform a statistical analysis. Moreover, *Cx32* mRNA levels in 5xFAD mice from both age groups exhibited significant induction compared with the related WT mice, which is a possible compensatory molecular mechanism in response to the loss of Cx32 GJs. In addition, Cx32 GJ plaques were not detectable in the vicinity of amyloid plaques in 12M 5xFAD GM, and we found reduced Cx32 immunoreactivity in the WM that corresponds to the uncrossed CST in rodents [70]. These findings indicate disrupted oligodendroglial connectivity in the form of intra-myelin and inter-oligodendrocyte Cx32–Cx32 GJs with unknown consequences for the overall spinal cord physiology. Loss of reflexive Cx32 GJs within myelin sheath limits the transport of ions and energy supply to the axon and disrupts the Ca^2+^ and K^+^ homeostasis and axon–glial communication, which enhances neuronal stress. This was previously shown in the GM of MS cases [4,54,66].

Additionally, we identified partial myelination deficits in the vicinity of amyloid plaques with weak immunoreactivity of myelin markers MBP and PLP, which coincided with the loss of Cx32 GJs. Although we found low immunoreactivity of the NF-H axonal marker around the boundaries of the plaques, immunoblot analysis did not show significant reduction at the protein level, indicating that the axons were largely preserved at those regions. Previous publications have demonstrated that the 5xFAD and APP/PS1 AD models display axonal pathology in the brain and spinal cord in the form of axonal swellings with APP aggregates, which might affect the propagation of signals causing motor decline [31,44,71,72,73]. Our results imply the selective loss of myelin Cx32 GJ plaques with an unclear effect. Future research should elucidate whether the disruption of reflexive Cx32 GJs could be a contributing factor for the reported myelin and axonal swelling or if it is a secondary pathological event.

In conclusion, we describe age-dependent alterations of glial GJs in the spinal cord of the 5xFAD mouse model of early-onset AD with marked memory and motor impairments. Our data provide evidence of the pathological expression pattern of oligodendroglial and related astroglial GJs in the cervical and lumbar GM of 12M 5xFAD mice, which correlate with age-dependent progressive Aβ-accumulation and age-dependent motor dysfunction. The establishment of increased A/A GJs and O/O GJ connectivity among glial cells in the spinal cord of this model might accelerate AD-related pathology and the development of motor dysfunction and dementia. Although various CNS pathological conditions could cause alteration of glial connexins, the neuroinflammation-driven upregulation of astroglial Cx43 and oligodendroglial Cx47 in the GM appears to be specific for the 5xFAD mouse spinal cord. The accumulated Aβ-driven GJ pathology in the pyramidal and extra-pyramidal pathways in the 5xFAD spinal cord and brain might be the molecular basis of the deficits in locomotor activity, motor coordination, and performance that were ascertained during behavioral analysis. It remains to be determined whether glial GJ changes are associated with the initiation or/and progression of AD-related neuropathology and sensorimotor dysfunction. Importantly, connexin-mediated chronic inflammation might be the target for novel therapeutic approaches to treat neurodegenerative diseases with shared connexin pathology in different brain and spinal cord regions.

## 4. Materials and Methods

### 4.1. Experimental Animals

Male 5xFAD (Tg6799) transgenic mice were purchased from Jackson Laboratory (Bar Harbor, ME USA 04609, www.jax.org, accessed on 1 October 2022) and crossed with B6/SJL F1 hybrid female mice. 5xFAD mice co-express the human transgenes APP and PSEN1 in high abundance under the influence of the neuronal-specific elements of the murine Thy-1-promoter. These transgenes have the following mutations: APP (Swedish: K670N/M671L, Florida: I716V, London: V717I mutations) and PSEN1 (M146L, L286V mutations) [32]. Animals were genotyped with polymerase chain reaction (PCR) prior to use and were kept in the specific pathogen-free mouse facility of the Cyprus Institute of Neurology and Genetics under standard controlled conditions of temperature (21–23 °C), humidity, air exchange, with and a routine 12 h light/dark cycle. Additionally, mice were housed up to five in each cage and provided with standardized mouse diet and water ad libitum. The offspring were divided into two age groups (3 and 12 months, represented as 3M and 12M). Transgenic mice (*n* = 6 per age group) were used in all experiments in parallel with age-matched wild-type (WT) mice (*n* = 6 per age group). Female mice were used for the IHC experiments, male mice for the immunoblot and qPCR experiments, and mice of both genders were used for the behavioral tests and showed no (sex-related) differences in their motor behavioral performance. All animal procedures were approved by the Cyprus Government’s Veterinary Services (Council Directive 86/609/EEC) as stated by the EU guidelines (CY/EXP/PRL01/2022).

### 4.2. Foot-Slip Test

Mice performed foot-slip test to assess motor coordination. During the procedure [74,75], mice were placed in a 15 cm × 15 cm × 15 cm plexiglass container with a metal wire floor that consisted of 1.25 cm spacing and with a 1.25 cm grid suspended 1.25 cm above the floor. Mice were kept in the box for 1 h before each session to be acclimated to the environment. The trial involved 50 steps. In case of a misstep, which resulted in the hind limb or forelimb falling through the grid, a score of one was assigned if the limb was withdrawn prior to touching the floor, and a score of two was given if the floor was touched by the limb. A video camera was used to film the mice, and video recordings were evaluated in slow motion.

### 4.3. Footprint Test

The footprint test was performed to assess the gait of mice. The mice underwent acclimatization to the environment for 15 min and were permitted to practice two runs, after which the hind-paws were painted with non-toxic blue dye [74,76]. Then, each mouse individually was placed at one end of the alleyway and was motivated to walk down and cross to the opposite side of the open-top narrowed runway covered with white paper on the floor. The length of the runway was 70 cm long with 16 cm width and 30 cm-high walls. Mice were given two trials to achieve a clear set of footprints that were visible and in a straight line. The duration of each trial last until the mice arrived at the opposite end of the box. Mice that stayed motionless for more than 10 min were withdrawn from the test and given one last trial. Once the footprints had dried, the distance between subsequent footprints from the same back paw (stride length) and the distance between hind-paws (stride width) were estimated non-digitally with a ruler.

### 4.4. Rotarod

Mice performed the rotarod test to examine balance and motor co-ordination [45,77]. Animals were placed on a 3.5 cm diameter rod, rotating at an accelerated speed from 0 to 40 rotations per minute (rpm). Mice underwent sessions of training consisted of six trials per day with 15-min rest periods between trials for three consecutive days. The test was completed once the mouse fell from the rod or after the mouse remained on the rod for up to six min and then was removed. Testing was performed on the fourth day, and after each trial, mice were kept in their cage for one minute before the next one. The latency to fall was recorded using a timer. Before every session, mice were acclimated within the test room for 60 min.

### 4.5. Hind-Limb Grip Strength Test

The strength of hind-limbs was estimated using the grip strength meter [45,78] according to the manufacturer’s instructions. During the test, mice were held by the tail towards the apparatus until the grid was grabbed by the hind paws. Then, mice were gently pulled back until they released the grid. The force (in grams; g) was calculated by the equipment. All force measurements were repeated five consecutive times, and the hind limb force was calculated by averaging the scores of each trial for each animal.

### 4.6. Tissue Processing

Mice were deeply anesthetized with intraperitoneal administration (Avertin, 5 g/100 μL) and transcardially perfused with ice-cold 0.9% saline. Then, spinal cords were isolated by laminectomy and transferred into 4% paraformaldehyde (PFA) for additional post-fixation and kept in 20% sucrose in phosphate buffer (0.1 M PB) overnight (O/N) to avoid the formation of freezing artifacts in tissues. The following day, spinal cords were divided into whole distinct cervical, thoracic, and lumbar segments that were identified by their fundamental anatomical features [79], embedded in Optimum Cutting Temperature (OCT) compound, settled in a dry-ice/acetone cooling bath, and stored at −80 °C. Transverse histological sections of 12 μm thickness were cryosectioned, mounted on several microscopic glass slides, and stored at −20 °C.

### 4.7. Fluorescence Immunohistochemistry

Spinal cord cross sections were permeabilized in cold acetone for 10 min, blocked at room temperature (RT) with blocking solution 5% bovine serum albumin (BSA) in PBS/Triton X-100 (0.5%), and incubated at 4 °C O/N with the following primary rabbit monoclonal antisera against Olig2 (Merck Millipore, Burlington, MA, USA, 1:500), Iba1 (Biocare Medical, Pacheco, CA, USA, 1:500), Cx43 and Cx30 (Thermo Fisher Scientific, Waltham, MA, USA, 1:500), Cx47 (Thermo Fisher Scientific, 1:500), Cx32 (Thermo Fisher Scientific, 1:100), NF-heavy (Thermo Fisher Scientific, 1:1000), PLP (Abcam, Cambridge, UK, 1:3000), and primary mouse monoclonal antibodies against NeuN (Merck Millipore 1:400), β-amyloid [(6E10), Covance, 1:400)], glial fibrillary acidic protein [(GFAP), Sigma-Aldrich, St. Louis, MO, USA), 1:400)], Cx47 (Invitrogen, Waltham, MA, USA, 1:200), RT97 (DHSB, Iowa City, USA 1:1000), and APC (CC-1, Merck Millipore, 1:50). The following day, spinal cord sections underwent washes with 1× PBS and were incubated for 1 h with the related secondary antibodies in blocking solution: goat anti-mouse FITC (Jackson ImmunoResearch, West Grove, PA, USA, 11:100) and goat anti-rabbit Alexa Fluor^®^ 594 (Jackson ImmunoResearch, 1:500). Sections were then counterstained with DAPI (Sigma-Aldrich) and mounted with Fluorescent Mounting Medium (DAKO). Stained histological sections were photographed with a Zeiss fluorescence microscope using a digital camera at 20×, 40×, and 60× magnification and the Zeiss Axiovision software ZEN 3.4 (Carl Zeiss Microimaging, Oberkochen, Germany). For accurate identification of the photographed cervical and lumbar sections, the atlas of the mouse spinal cord was used [80]. ImageJ software (Version 1.53c, Wayne Rasband, National Institutes of Health, Bethesda, MD, USA, http://imagej.nih.gov/ij, accessed on 1 October 2022) was used to analyze the photographed images (*n*= *3* images of the whole GM area per spinal cord tissue*/*per mouse, *n* = 6 mice per age group, *n* = 6 mice per genotype).

### 4.8. Analysis of the Degree of Neuroinflammation in the 5xFAD Spinal Cord by Immunostaining

In WT, 3M, and 12M 5xFAD mice, the degree of inflammation at the GM of cervical and lumbar spinal cord sections was estimated from GFAP/Iba1 double labeling/immunostaining images. The method of the analysis was obtained from [39,81], and ImageJ software was used. To assess the degree of astrogliosis and the activation of microglia, we measured the total GM area cover by GFAP and Iba1 immunofluorescence, and results were presented as percentage of total image area.

### 4.9. Semi-Quantification of Fluorescence Intensity of Cx43 and Cx30 at the Sites of Aβ Plaques

In 3M and 12M 5xFAD mice, the fluorescence intensity of Cx43 and Cx30 GJs at the level of Aβ plaques was estimated from Aβ/Cx43 and Aβ/Cx30 double immunostaining images of specific cervical and lumbar GM spinal cord histological sections. Beta-amyloid plaques found close to blood vessels were not included in the analysis. The method of the analysis was obtained from [25] using ImageJ software. For each Aβ plaque, three ellipses (with 60 rotations from each other) that encompass the whole perimeter and the core of the plaque were designed. The mean fluorescence intensity of Cx43 and Cx30 GJ plaques was measured in the three oval selected sections at the level of each Aβ plaque and in five areas away from Aβ plaques within the same image. Aβ plaques present at the spinal WM were not taken into account in the measurements. Additionally, the mean Cx fluorescence intensity at the level of Aβ plaques in 5xFAD mice was compared with the fluorescence intensity in five areas away from Aβ plaques and with the Cx fluorescence intensity in WT mice (five different areas were measured per image in WT mice).

### 4.10. Semi-Quantification of Fluorescence Intensity of Cx47 Plaque Formation near the Sites of Aβ Plaques

Gap junction plaques formed by oligodendrocytic Cx47 were interpreted as individual concentrations of connexin immunoreactivity around oligodendrocyte cell bodies. CC1^+^ mature oligodendrocytes with Cx47-positive puncta were selected within squares of 26.858 mm^2^, and the fluorescence intensity was measured after removing the background. Both Cx47 immunoreactivity and the number of CC1^+^ oligodendrocytes expressing Cx47-positive GJ plaques were measured in total GM using ImageJ software. CC1^+^ oligodendrocytes expressing Cx47-positive GJ plaques were not found within the location of the demyelinated lesions at the sites of the Aβ plaques. Data for analysis was obtained from CC1^+^/Cx47 double immunostaining images of specific cervical and lumbar GM histological sections of 3M and 12M 5xFAD mice and compared to WT controls.

### 4.11. Immunoblot Analysis

Fresh whole cervical and lumbar spinal cords were isolated by laminectomy, separated into two distinct tissue blocks based on their fundamental anatomical features, and were homogenized in separate tubes in 70 μL ice-cold RIPA buffer (1% NP-40, 0.5% C24H39Na04, 0.1% SDS, 2 mM EDTA), dissolved in 1× PBS along with protease and phosphatase inhibitors (Roche, Basel, Switzerland), sonicated on ice, and centrifuged for 30 min at 4 °C. Protein concentration was estimated with NanoDrop™ ND_100 spectrophotometer. Proteins (130 μg) from tissue lysates were fractioned by 12% SDS-PAGE [ddH20, 30% acrylamide-bis (29:1), 1.5M Tris-CL pH 8.8, 10% SDS, 10% APS, Temed] and transferred to an activated Hybond PVDF blotting membrane (GE Healthcare Life Sciences) using Trans-Blot Turbo Transfer System via the semi-dry method (Bio-Rad Laboratories Inc., Hercules, CA, USA). The immunoblots were blocked in 5% BSA in PBS containing 0.1% Tween-20 for 1 h at RT and then incubated O/N at 4 °C with mouse primary antiserum against Cx43 (Millipore, 1:1000), Cx32 (Thermo Fisher Scientific, 1:500), RT97 (DSHB, 1:1000), or β-Tubulin E7 (DSHB, 1:4000) or with rabbit primary antisera against Cx30 (Thermo Fisher Scientific, 1:500), Cx47 (Life Technologies 1:1000), MBP (Thermo Fisher Scientific, 1:5000), or NF-H (Abcam, 1:10,000). After 15 min of washes in PBS-T, membranes were incubated for 1 h at RT with a goat anti-mouse and goat anti-rabbit HRP-conjugated secondary antibody in blocking solution (Jackson Immunoresearch, 1:3000). Membranes were developed, and the bound antibodies were visualized by enhanced chemiluminescence system (ECL, GE Healthcare Life Sciences). Protein expression was quantified via optic band intensity estimation using ImageJ. For the quantification analysis, optic density (OD) levels of each protein of interest were normalized to the β-tubulin or GAPDH optic density bands, which were used as loading controls. All immunoblots were performed at least two times obtaining similar results.

### 4.12. RNA Extraction and Quantitative Real-Time PCR

Total RNA was extracted from whole cervical and lumbar spinal cord tissue blocks with RNeasy Lipid Tissue Mini kit (QIAGEN, Hilden, Germany) following the manufacturer’s protocol, and tissues were homogenized using 300 μL of QIAzol lysis buffer. Additionally, 200 μL of Chloroform was used to denature all the proteins. After DNase digestion, total RNA was quantified with a NanoDrop™ ND_100 spectrophotometer. Aliquots of 200 ng of total RNA from each sample were subjected to reverse transcription (RT)-PCR (25 °C for 10 min, 48 °C for 30 min, and 95 °C for 5 min) using TaqMan^®^ Reverse Transcription Reagents with final reaction volume 40 μL (Applied Biosystems). The total mRNA expression levels of *Gja1* (Cx43*), Gjb6* (Cx30), *Gjc2* (Cx47), and *Gjb1* (Cx32) were assessed by quantitative real-time PCR and compared to the expression of the housekeeping endogenous control gene *Tubb4a* (β-tubulin) using the following TaqMan Gene Expression assays: Cx43: Mm01179639_m1, Cx30: Mm00433661_s1, Cx47: Mm00519131-s1, Cx32: Mm01950058-s1, and Tubulin: Mm00726185_s1. Each sample was loaded in triplicate and contained 100 ng of cDNA, 1 μL of TaqMan^®^ Gene Expression Assay, and 10 μL of TaqMan^®^ Gene Expression Master Mix (end volume 20 μL). Cycle thresholds (Cts) of genes of interest were normalized against β-tubulin, and mRNA levels were calculated in 5xFAD and WT mice and presented as fold induction values (2^−ΔΔCt^) compared with WT control mice.

### 4.13. Statistical Analysis

Statistical analysis was performed using GraphPad Prism for Windows (Version 9.01, GraphPad Software, San Diego, CA, USA, www.graphpad.com, accessed on 1 October 2022). All data sets were evaluated for normality using the Shapiro–Wilk normality test. For comparison of means between three or more independent groups, one-way ANOVA followed by a Kruskal–Wallis multiple comparisons test were performed. All data sets were expressed as the mean ± standard deviation (SD), and any *p*-values of *p* < 0.05 were considered statistically significant. Details showing which statistical test was performed in each experiment are reported in the figure legends.

## Figures and Tables

**Figure 1 ijms-23-15597-f001:**
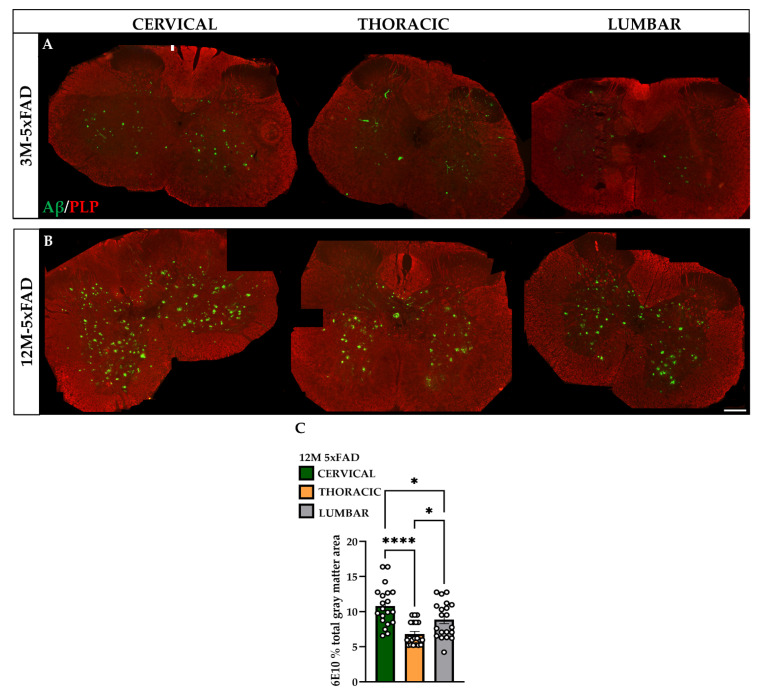
Age-related progressive Aβ accumulation in the spinal cord of 12M 5xFAD mice. (**A**,**B**) Spinal cord cross-sections (12 µm thick) of 3M and 12M 5xFAD mice stained with 6E10 (green) primary antibody revealed individual plaques (green dots) in the cervical (C2), thoracic (T2), and lumbar (L2) GM sections that progressively increased with age in 12M compared with 3M 5xFAD mice. (**C**) Quantification of the percentage of the total GM area occupied by 6E10 antibody showed the highest deposition of Aβ senile plaques in the cervical and lumbar sections of 12M 5xFAD mice compared with thoracic sections. Statistical analysis was performed by one-way ANOVA followed by Kruskal–Wallis multiple comparisons test (*n* = 6 for all genotype and age groups). Data are presented as mean ± SD. Scale bars = 100 µm. Significance is given as: * *p* < 0.05, **** *p* < 0.0001.

**Figure 2 ijms-23-15597-f002:**
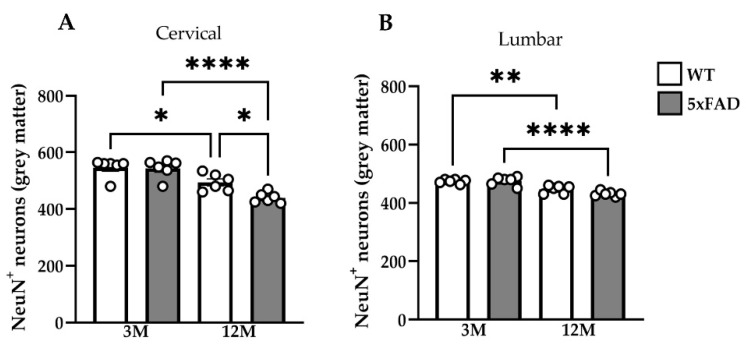
Age-related neuronal loss in the spinal cord of 12M 5xFAD mice. Quantification of the mean number of NeuN^+^ neurons in cervical C2 (**A**) and lumbar L2 (**B**) sections displayed a significant neuronal loss in 12M 5xFAD mice compared with their WT littermates in cervical (**A**) but not in lumbar (**B**) sections as well as compared with 3M 5xFAD mice in cervical and lumbar GM. Statistical analysis was performed by one-way ANOVA followed by Kruskal–Wallis multiple comparisons test (*n* = 6 for all genotype and age groups). Data are presented as mean ± SD. Significance is given as: * *p* < 0.05, ** *p* < 0.01, **** *p* < 0.0001.

**Figure 3 ijms-23-15597-f003:**
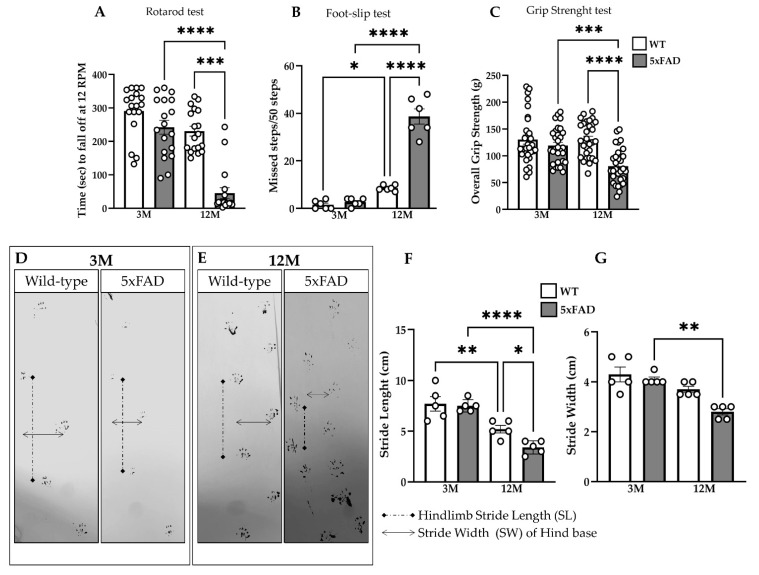
Decline of motor function in 12M 5xFAD mice. (**A**) Rotarod testing at 12 rotations per minute (rpm) showed significantly worse performance of 12M 5xFAD compared with WT mice of the same age and with the 3M 5xFAD mice. (**B**) Foot-slip test showed more missed steps in 12M 5xFAD compared with WT littermates and 3M 5xFAD mice. (**C**) Grip strength evaluation revealed significantly reduced strength in old 5xFAD mice compared with the WT group and with the 3M 5xFAD mice. (**D**,**E**) Representative photographs of footprints from 5xFAD and WT mice of 3 and 12 months of age. Quantitative footprint analysis revealed that 12M 5xFAD mice had an overall more severe gait phenotype resulting from reduced stride length (**F**) and stride width (**G**) of hind paws compared with WT animals of the same age. Statistical analysis was performed by one-way ANOVA followed by Kruskal–Wallis multiple comparisons for all the behavioral tests (*n* = 6 for all genotype and age groups). The same 3M and 12M mice were used for all the experiments. Data are presented as mean ± SD. Significance is given as: * *p* < 0.05, ** *p* < 0.01, *** *p* < 0.001, **** *p* < 0.0001.

**Figure 4 ijms-23-15597-f004:**
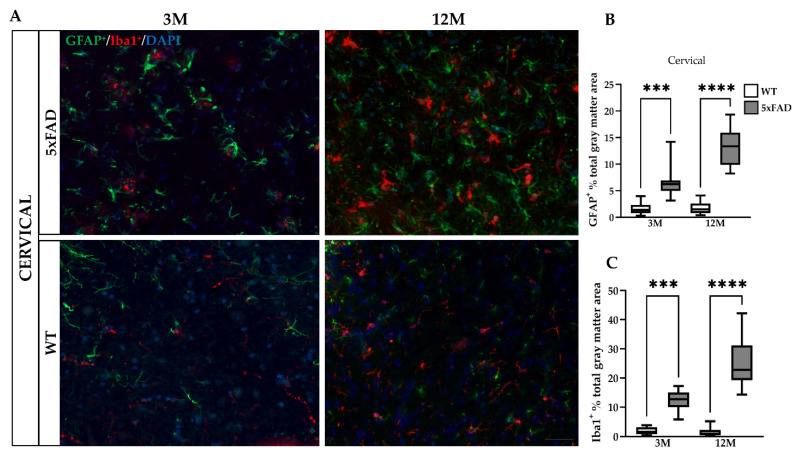
Severe astrogliosis and microglia activation in the spinal cord of 5xFAD mice. (**A**) In cervical C2 and (**D**) lumbar L2 spinal cord GM from 5xFAD and WT mice, double immunofluorescence staining was performed with astrocytic marker GFAP (green) and microglial marker Iba1 (red). 5xFAD mice showed intense astrogliosis in the spinal cord at 12 months of age and to a lesser degree also at 3 months, whereas WT mice of both ages showed no astrogliosis. Quantification of the percentage of the total GM area stained by GFAP^+^ reactive astrocytes (**B**,**E**) and Iba1^+^ activated microglia (**C**,**F**) at both spinal cord levels displayed significantly increased immunoreactivity, which confirmed progressive neuroinflammation in 5xFAD mice. The statistical analysis was performed by one-way ANOVA followed by Kruskal–Wallis multiple comparisons test (*n* = 6 for all genotype and age groups). Data are presented as mean ± SD. Scale bars = 50 µm in (**A**,**D**). Significance is given as: *** *p* < 0.001, **** *p* < 0.0001.

**Figure 5 ijms-23-15597-f005:**
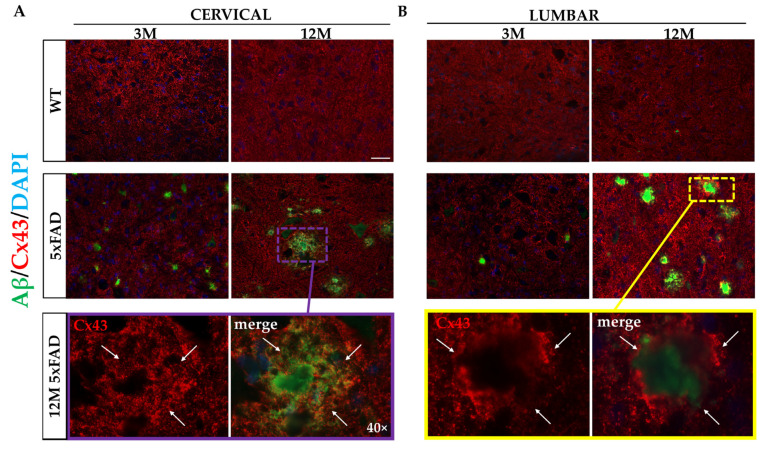
Increased immunoreactivity of astrocytic Cx43 and Cx30 around β-amyloid plaques within the 5xFAD spinal cord. (**A**,**C**) In cervical C3 and (**B**,**D**) lumbar L3 spinal cord sections from 5xFAD and WT mice, double immunofluorescence staining was performed with β-amyloid antibody 6E10 (green) and astrocytic Cx43 (**A**,**B**) or Cx30 (**C**,**D**) antibodies (red). In 12M 5xFAD mice, the immunoreactivity of both astrocytic connexins around the perimeter of amyloid plaques in the GM was increased, as can be observed with white arrows in the higher magnification insets highlighted in purple rectangles for cervical sections (**A**,**C**) and yellow rectangles (**B**,**D**) for lumbar sections. Quantification of the fluorescent intensity of Cx43 (**E**,**F**) and Cx30 (**G**,**H**) in GM areas around amyloid plaques, areas off plaques, and in corresponding WT areas confirmed that the immunoreactivity of Cx43 and Cx30 was significantly higher around the perimeter of Aβ plaques in comparison with areas off plaques in 5xFAD mice or with WT areas. Statistical analysis was performed by one-way ANOVA followed by Kruskal–Wallis multiple comparisons test (*n* = 6 for all genotype and age groups). Data are presented as mean ± SD. Scale bars = 50 µm; magnified view scale bar = 10 µm (**A**–**D**). Significance is given as: ** *p* < 0.01, *** *p* < 0.001, **** *p* < 0.0001.

**Figure 6 ijms-23-15597-f006:**
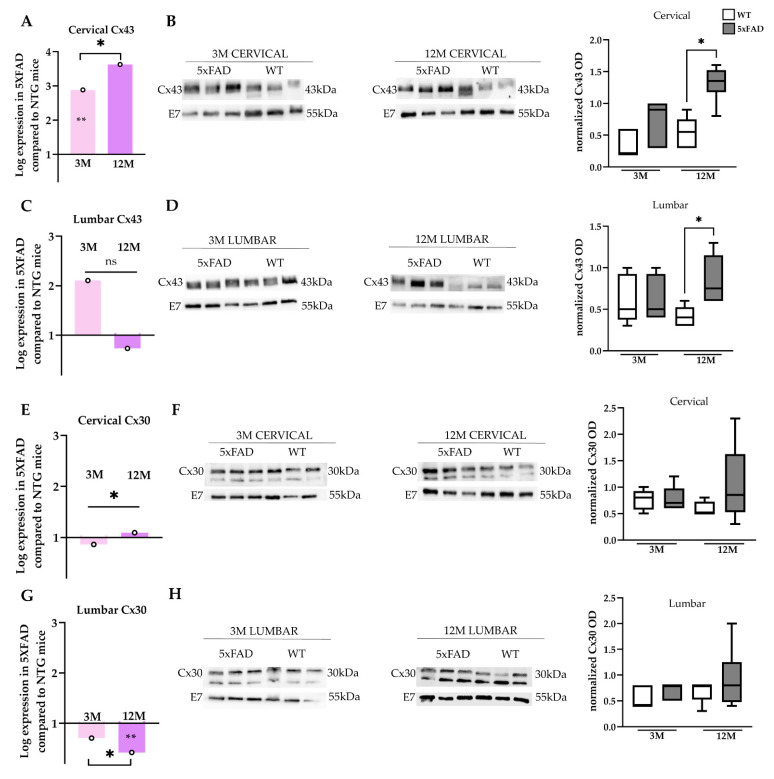
*Cx43* and *Cx30* mRNA and protein levels in the spinal cord of 5xFAD mice. RT-PCR analysis of *Cx43* (**A**,**C**) and *Cx30* (**E**,**G**) mRNA transcript levels in the cervical and lumbar spinal cord tissue of 3- and 12-month-old 5xFAD and WT mice. The asterisks inside the columns of the mRNA graphs denote the *p*-values representing significance of connexin levels in 5xFAD compared with WT controls, while the asterisks outside the columns show the significance between different age groups of 5xFAD mice. Quantification of normalized band optic density (OD) revealed increased Cx43 protein levels (**B**,**D**) and unchanged Cx30 protein levels (**F**,**H**) in 12M 5xFAD mice compared with the WT controls at both spinal cord levels. *β*-Tubulin was used as an internal control for the mRNA analysis and as a loading control for the immunoblot analysis (E7). Statistical analysis for the mRNA was performed by unpaired *t*-test (*n* = 6 for all genotype and age groups) and for the immunoblots was performed by one-way ANOVA followed by Kruskal–Wallis multiple comparisons test (*n* = 6 for all genotype and age groups). Data are presented as mean ± SD. Significance is given as: * *p* < 0.05, ** *p* < 0.01.

**Figure 7 ijms-23-15597-f007:**
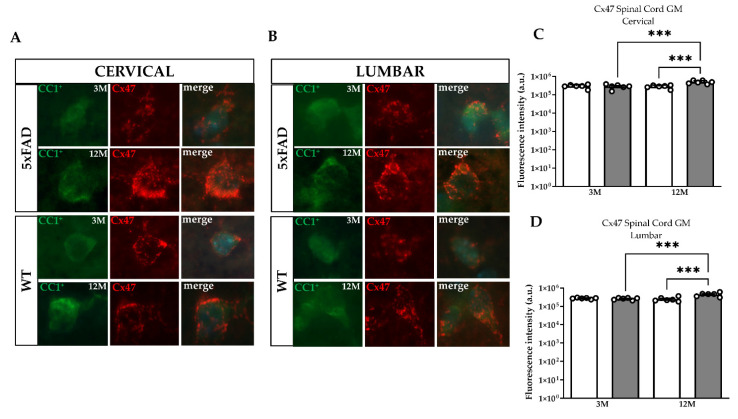
Increased immunoreactivity of oligodendroglial Cx47 expressed by mature oligodendrocytes within the GM of the 5xFAD spinal cord. Double immunofluorescence staining was performed (**A**) in cervical C3 and (**B**) lumbar L3 spinal cord segments from 5xFAD and WT mice using a marker for mature oligodendrocytes CC1 (green) and oligodendroglial anti-Cx47 antibody (red). Cell nuclei were counterstained with DAPI (blue). (**C**,**D**) Quantification of the fluorescent intensity of Cx47-positive GJ plaques in the GM of both spinal cord sections in 5xFAD and WT mice shows that immunoreactivity levels of Cx47 were significantly increased in 12M 5xFAD mice compared with the WT group and with the 3M 5xFAD group. Quantification of the mean number of CC1^+^ mature oligodendrocytes (**E**,**F**) expressing Cx47-positive GJ plaques in the GM of 5xFAD and WT mice of all ages showed increased numbers of CC1^+^ in 12M 5xFAD mice in both spinal cord sections in comparison with their WT littermates. The statistical analysis was performed by one-way ANOVA followed by Kruskal–Wallis multiple comparisons test (*n* = 6 for all genotype and age groups). Data are presented as mean ± SD. (**A**,**B**) Scale bars = 10 µm. Significance is given as: ** *p* < 0.01, *** *p* < 0.001, **** *p* < 0.0001.

**Figure 8 ijms-23-15597-f008:**
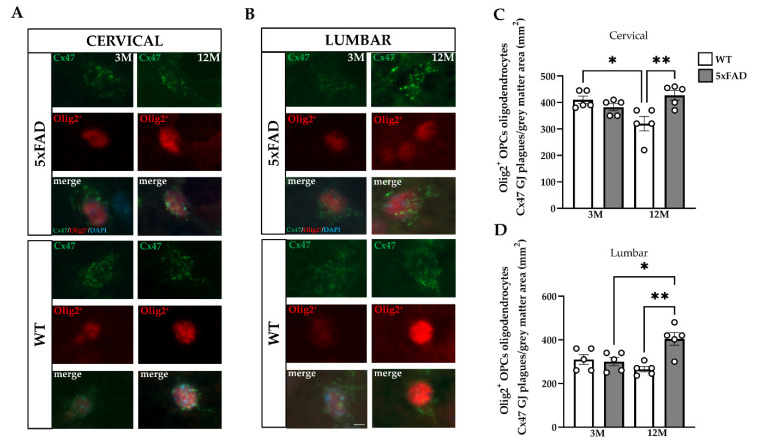
Elevation of oligodendrocyte precursor cells (OPCs) expressing Cx47 within the GM of the 5xFAD spinal cord. Double immunofluorescence staining was performed (**A**) in cervical C3 and (**B**) lumbar L3 spinal cord segments from 5xFAD and WT mice using a marker for oligodendrocyte precursors (OPCs), Olig2 (red), and anti-Cx47 antibody (green). Cell nuclei were counterstained with DAPI (blue). Quantification of the mean number of Olig2^+^ precursors (**C**,**D**) expressing Cx47-positive GJ plaques in the GM of 5xFAD and WT mice of all ages showed increased numbers of Olig2^+^ OPCs in 12M 5xFAD mice in both spinal cord sections in comparison with their WT littermates. The statistical analysis was performed by one-way ANOVA followed by Kruskal–Wallis multiple comparisons test (*n* = 6 for all genotype and age groups). Data are presented as mean ± SD. (**A**,**B**) Scale bars = 10 µm. Significance is given as: * *p* < 0.05, ** *p* < 0.01.

**Figure 9 ijms-23-15597-f009:**
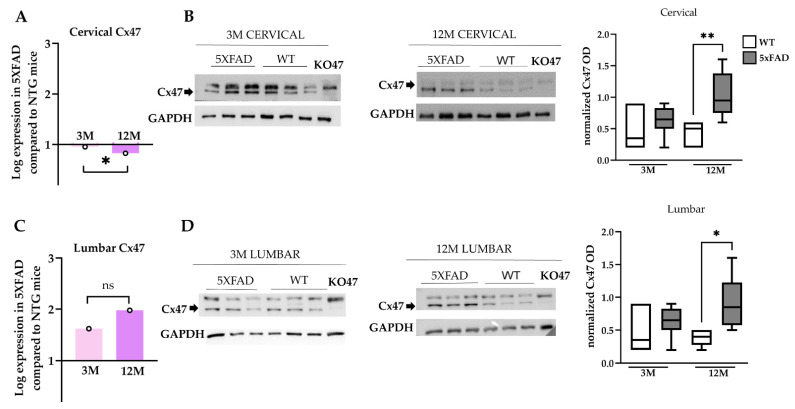
*Cx47* mRNA and protein levels in the spinal cord of 5xFAD mice. Real-time PCR of *Cx47* mRNA transcript levels in cervical (**A**) and lumbar (**C**) spinal cord sections of 3M and 12M 5xFAD and WT mice. The asterisks inside the columns of mRNA graphs indicate the *p*-values representing significance of connexin levels in 5xFAD compared to WT controls, while the asterisks outside the columns show the significance between different age groups of 5xFAD mice. (**B**,**D**) Quantification of normalized band optic density (OD) revealed increased Cx47 protein levels in 12M 5xFAD mice compared with the WT control group in both spinal cord sections. *β*-Tubulin was used as an internal control for the mRNA analysis, and GAPDH was used as a loading control for the immunoblot analysis. Statistical analysis for the mRNA was performed by unpaired *t*-test and for the immunoblots was performed by one-way ANOVA followed by Kruskal–Wallis multiple comparisons test (*n* = 6 for all genotype and age groups). Data are presented as mean ± SD. Significance is given as: * *p* < 0.05, ** *p* < 0.01.

**Figure 10 ijms-23-15597-f010:**
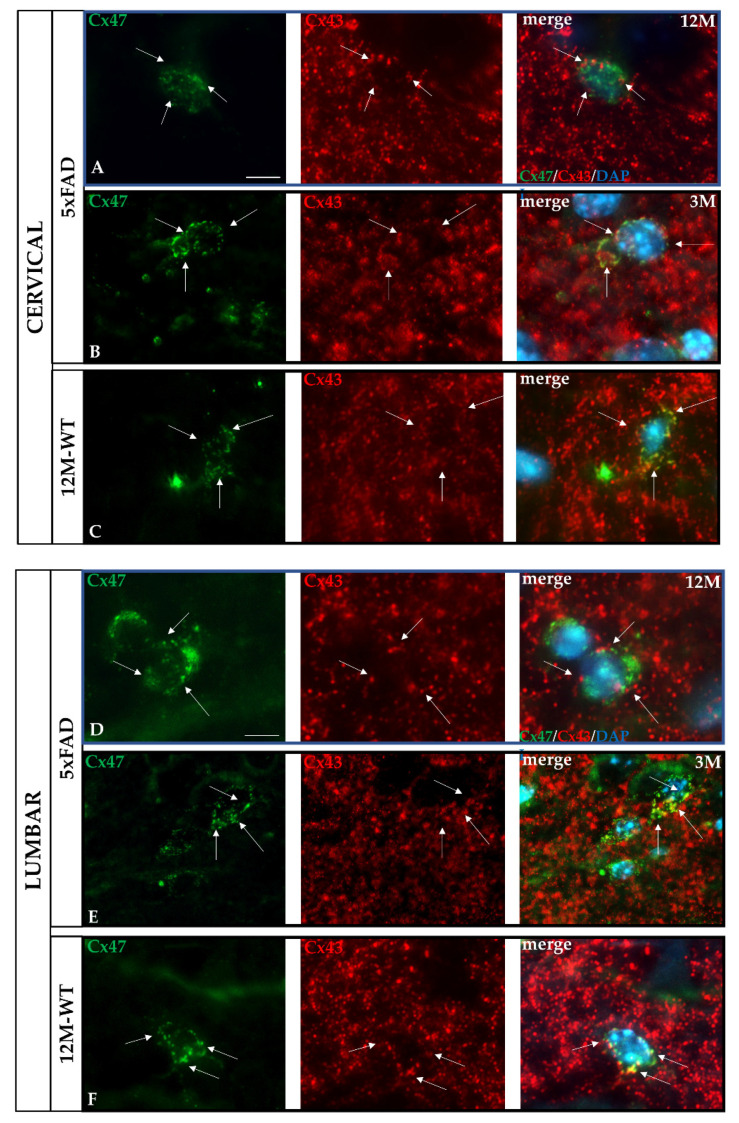
Partial disruption of heterotypic oligodendrocyte/astrocyte (O/A) GJs in the spinal cord of 12M 5xFAD mice. (**A**–**F**) Double immunostaining using anti-Cx47 (green) and anti-Cx43 (red) antibody. Cell nuclei were stained with DAPI (blue). In cervical C3 and lumbar L3 GM, 12M WT mice (**C**,**F**) exhibited normal colocalization of Cx43 with Cx47 around oligodendrocyte cell bodies. (**A**,**D**) However, in GM of 12M 5xFAD mice, there was an evident increase in Cx43 GJ plaques that do not colocalize (white arrows) with Cx47. Cx47 GJ plaques also appeared increased in oligodendrocytes of 5xFAD compared with WT mice. Magnified view scale bar = 10 µm.

**Figure 11 ijms-23-15597-f011:**
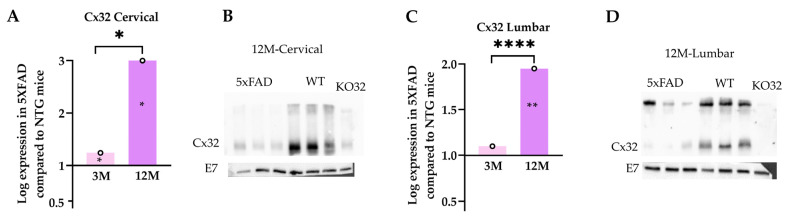
*Cx32* mRNA and protein levels in the spinal cord of 5xFAD mice. *Cx32* mRNA levels increased significantly in the cervical (**A**) and lumbar (**C**) spinal cord tissues at both ages of 5xFAD mice. The asterisks inside the columns of mRNA graphs indicate *p*-values representing significance of connexin levels in 5xFAD compared with WT controls, while the asterisks outside the columns show the significance between different age groups of 5xFAD mice. (**B**,**D**) However, immunoblot analysis revealed reduced Cx32 protein levels in 12M 5xFAD mice compared with the WT samples at both spinal cord levels. *β*-Tubulin was used as an internal control for the mRNA analysis and as a loading control (E7) for the immunoblot analysis. The statistical analysis for the mRNA was performed by unpaired *t*-test (*n* = 6 for all genotype and age groups). Data are presented as mean ± SD. Significance is given as: * *p* < 0.05, ** *p* < 0.01, **** *p* < 0.0001.

## Data Availability

Not applicable.

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
