# Peer review of "Glial Gap Junction Pathology in the Spinal Cord of the 5xFAD Mouse Model of Early-Onset Alzheimer’s Disease"

_ijms, 2022, doi:10.3390/ijms232415597_

Round 1

Reviewer 1 Report

The manuscript titled: "Glial gap junction pathology in the spinal cord of the 5xFAD 2 mouse model of early-onset Alzheimer’s disease" by Pechlivanidou et al is important and interesting. However, further updates might be needed.

Minor comments

  1.  The authors mentioned the PSEN1 gene's importance in their introduction and the discussion, however, it is not clear how this gene is directly connected to their study.
  2. Figure 1 is missing the control.
  3. In Figure 5a, apparently the magnification of the 12 M is shown under the 3 Month title, which could be confusing for the readers.
  4. There might not be any pictures showing the CX32 expression localization, although that seems to be the main aim of the study.

Major comments.

The main hindrance to reading the manuscript is the lack of focus.

It is not clear whether the authors' main aim is to show the expression of certain connexins in relation to Alzheimer's or to measure the effect of Alzheimer's on mice's gait. They seem to be two different aspects. Is there a relationship between the mice's gait and the connexin investigated? 

For example, the authors say that

"Quantitative footprint analysis revealed that 12M 5xFAD mice had overall a 217 more severe gait phenotype resulting from reduced stride length". That is a very good observation, but how it is connected to connexin expression is not yet clear. 

Another example is that in the introduction, the authors say that the expression of CX43/CX43 was found to be in A/A. However, their main aim seems to be to localize CX43. A rewrite of the introduction and the discussion section could be needed to clarify this point.

Overall, the manuscript may be fragmented. Although effort has been put toward compiling the results, writing the "story" and connecting it to the main goal might need a further update.

Author Response

Thank you.

Reviewer 2 Report

In the article titled “Glial gap junction pathology in the spinal cord of the 5xFAD mouse model of early-onset Alzheimer’s disease” the main goal of Pechlivanidou et al., was to evaluate possible alternations in the expression of glial GJ proteins and their association with the progressive accumulation of Aβ-plaque pathology within the spinal cord of the 5xFAD model of early-onset AD. They present novel findings of Aβ-induced increased expression of astrocytic Cx43 and its oligodendrocytic partner Cx47 accompanied by proliferation of oligodendrocyte linage cells and a reduced expression of GJ protein Cx32 in the cervical and lumbar gray matter of 12M 5xFAD model.

As far as I'm concerned, the work done so far is very good and I was very happy for having had the opportunity to read this article which I find well written and whose conclusions can be very interesting.

But I have anyway two minor requests for the authors:

Materials and Methods

4.1. Experimental Animals

Line 1060:  Female mice were used for the IHC experiments, male mice for the immunoblot and qPCR experiments and mice of both genders were used for the behavioral tests as showed no (sex-related) differences in their motor behavioral performance.

Have the same tests been made, to which males were subjected, to females and vice versa? If yes, what results have they given?

Statistical Analysis

Is it possible to show the data in the graph through the use of boxplot? I think will be better to understand the distribution of the data and underline the major differences.

Author Response

Thank you.

Reviewer 3 Report

This is an interesting mouse study into AD-associated motor dysfunction mechanisms in the spinal cord.

These are my main comments on the presented study:

1. There is a recognized differential course of pathology development between male and female 5xFAD mice; i.e. the males lag behind females. In addition, there is strong evidence that neuroinflammation appears to be driven by different cells/mechanisms in the two sexes. As such, it is always difficult to properly align the two sexes and generalize from one to both. In this study m and f mice were used for different tests. Why did the authors not use BOTH sexes in each test OR one sex across the board. While the use of both sexes is commendable, this approach is not really achieving this and it does raise uncertainty about how well the different tests can be aligned.

2. The authors collect cervical and thoracic spinal cords but the levels and anatomical markers for these are not indicated. In addition, it is not entirely clear how histological sections were obtained - while the authors indicate 18 sections of 12um thickness, they later 6 images and I am not entirely clear how this works. In addition, how are the 18 images selected during sectioning as the sections are quite sizeable and would involve many more sections. In short, I would like to have more information about how representative the analyzed sections are - i.e. a very clear and explicit explanation of anatomical region --> section --> sampling for analysis.

3. I have similar questions about level selection and identification for the immunoblot and RNA - were they the whole cord section or a selected section? What were the anatomical markers for the levels?

4. The authors used GFAP to quantify reactive astrocytes. GFAP is tricky as it stains progenitors and has complex reactions to stimuli - GFAP is not necessarily proportional to the stimulus and with chronic ongoing trauma it has been shown to be reduced at times. There are myriad other markers that serve to specifically identify astrocytes and a reactive status. A little more information on the use of morphology (presumably) to identify astrocytes in methods would be useful to help us understand the validity of the approach.

5. Related to the above, a quantification of microglial phenotypes would be interesting in this context. In other words, quantifying numbers is one aspect but a subcategorizing of microglial types based on morphology would provide more useful information about functional status. Microglia activation isn't really demonstrated by Iba1 staining but by morphological and functional changes which I do not see.

6. I would vastly prefer to see the bar plots accompanied by individual dot plots to demonstrate sample size per test AND to clearly demonstrate the group separation and distribution of individuals. This is much more informative and I would strongly recommend that the authors provide this. In addition, this will allow us to ensure that the samples are balanced and that there was no loss of samples or unexpected and unexplained exclusions of data.

7. There are minor grammatical errors throughout, attention to which would improve the legibility of the paper.

Thank you.

Author Response

Thank you.

Round 2

Reviewer 1 Report

Thank you for addressing my comments.